# RetrievalAttention: Accelerating Long-Context LLM Inference via Vector Retrieval

Di Liu[1,2][†] Meng Chen[1,3][†] Baotong Lu[1][§] Huiqiang Jiang[1], Zhenhua Han[1], Qianxi Zhang[1]
Qi Chen[1], Chengruidong Zhang[1], Bailu Ding[1], Kai Zhang[3], Chen Chen[2][§] Fan Yang[1]
Yuqing Yang[1], Lili Qiu[1]

[1]Microsoft Research    [2]Shanghai Jiao Tong University    [3]Fudan University

[1]{baotonglu, hjiang, qiazh, cheqi, fanyang, yuqyang}@microsoft.com
[2]{liu-di, chen-chen}@sjtu.edu.cn
[3]mengchen22@m.fudan.edu.cn, zhangk@fudan.edu.cn

## Abstract

Transformer-based Large Language Models (LLMs) have become increasingly important. However, scaling LLMs to longer contexts incurs slow inference speed and high GPU memory consumption for caching key-value (KV) vectors. This paper presents RetrievalAttention, a training-free approach to both accelerate the decoding phase and reduce GPU memory consumption by pre-building KV vector indexes for fixed contexts and maintaining them in CPU memory for efficient retrieval. Unlike conventional KV cache methods, RetrievalAttention integrate approximate nearest neighbor search (ANNS) indexes into attention computation. We observe that off-the-shelf ANNS techniques often fail due to the out-of-distribution (OOD) nature of query and key vectors in attention mechanisms. RetrievalAttention overcomes this with an attention-aware vector index. Our evaluation shows RetrievalAttention achieves near full attention accuracy while accessing only 1-3% of the data, significantly reducing inference costs. Remarkably, RetrievalAttention enables LLMs with 8B parameters to handle 128K tokens on a single NVIDIA RTX4090 (24GB), achieving a decoding speed of 0.107 seconds per token.

## 1   Introduction

The advancements of transformer-based large language models (LLMs) [1] have shown exceptional capabilities in processing long contexts, with support up to 10 million tokens per context [2, 3]. This capability of analyzing extensive data facilitates numerous downstream applications, including document-based question answering (QA) [4], repository-level code analysis and generation [5], and personalized AI agents [6]. However, the use of long contexts poses challenges in inference efficiency, as both the size of the key-value (KV) cache and inference latency for KV accesses increase linearly with the context length. Consequently, how to reduce storage costs and token accesses are crucial for enhancing long-context inference efficiency, particularly in resource-constrained environments such as local deployments [7] or edge devices [8].

Recent studies [9, 10, 11] reveal that in many long-context applications, such as document question answering (QA) and in-context learning, a significant portion of the context remains constant and is reused across multiple requests. For instance, when conducting QA on a corpus, these documents are introduced to the LLM as a prefix and then re-utilized to answer different questions (i.e., user queries). As another example, some fixed contexts containing domain-specific knowledge are used to

---

[†]Work performed during the internship at Microsoft Research.
[§]Corresponding authors: Baotong Lu and Chen Chen.

39th Conference on Neural Information Processing Systems (NeurIPS 2025).

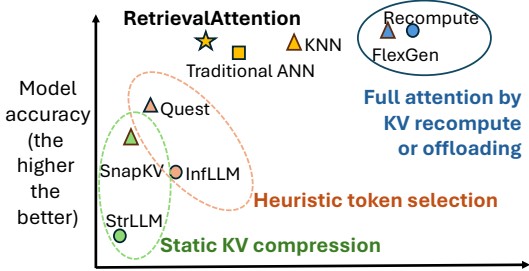

Figure 1: RetrievalAttention achieves similar task accuracy as full attention but exhibits extremely low decoding latency.

| Prompt Length | 128K | 256K | 512K | 1M |
|---|---|---|---|---|
| Total Latency (s) | 32.8 | 111 | 465 | 1,765 |
| FFN (s) | 7.6 | 15 | 31 | 70 |
| Attention (s) | 25.2 | 96 | 434 | 1,695 |
| GPU Memory KV Cache (GB) | 15.6 | 31.2 | 62.5 | 125 |

Table 1: Decoding latency and memory required for KV cache of Llama-3-8B across different context lengths on one A100 GPU.

supplement system or user prompts, improving generation quality. These observations have motivated existing research [9, 10] to pre-process fixed contexts for generating KV caches and reusing them in subsequent requests. This approach avoids redundant computation and thus enhances the efficiency of long-context inference, particularly by reducing the time to generate the first token.

However, the generation of new tokens during the decoding phase still suffers from inherent limitations of the conventional KV cache both in GPU memory consumption and inference latency, as all tokens from the cached context still participate in attention computation in each decoding step. A promising solution lies in leveraging the dynamic sparsity inherent in the attention mechanism [12], which allows for selectively attending to a subset of tokens that are critical to the current query vector. However, existing studies typically rely on static [13, 14] or heuristic [15, 16, 17] token selection methods, which often result in inaccurate identification and a significant drop in model accuracy.

We propose integrating Approximate Nearest Neighbor Search (ANNS) [18] into attention computation for the fixed contexts to enhance the inference efficiency. Our key insight stems from the observation of the mathematical equivalence between sparse attention computation and maximum inner product search[1]. Specifically, searching the ANNS index built for key vectors can return those with the highest inner product values in sublinear time. The retrieved key vectors are precisely aligned with the critical tokens required by the sparse attention mechanism, resulting in higher approximation accuracy compared to previous identification methods, as illustrated in Figure 1. The fixed contexts allow for building the indexes offline and reusing these indexes across requests when user queries come. Moreover, most ANNS algorithms are compatible with CPU implementations, enabling strategic allocation of GPU and CPU memory, which facilitates long-context inference on devices with limited GPU memory.

Leveraging ANNS for attention mechanism presents a unique challenge: the out-of-distribution (OOD) problem between query and key vectors. Most ANNS engines operate under the assumption that both query and key vectors are drawn from the same data distribution. However, this assumption does not hold in this context due to the different projection weights for query and key vectors in attention mechanism. Our measurement study shows that query vectors deviate more than $10\times$ farther from key vectors compared to that between in-distribution data. Unfortunately, the effectiveness of ANNS degrades significantly under the OOD problem. In particular, our empirical analysis indicates that maintaining a high inference accuracy requires conventional ANNS scanning more than 30% of all key vectors to identify the critical ones. This fails to fully leverage the inherent sparsity of attention and nullifies the ANNS efficiency benefits. To the best of our knowledge, we are the first to identify the challenge of OOD in using ANNS indexes to retrieve important tokens for sparse attention computation — a factor that is crucial for inference efficiency and accuracy.

In this work, we present RetrievalAttention, a long-context inference method that leverages ANNS indexes for efficient dynamic sparse attention in scenarios or applications where the fixed context takes a large proportion of prompt. To address the challenge of OOD, RetrievalAttention proposes a vector index tailored for the attention mechanism to mitigate the distribution gap between query and

---

[1]Maximum inner product search can be viewed as similarity search and efficiently solved by ANNS indexes [18]

key vectors. This approach allows for the traversal of only a small subset of key vectors (1–3%) to identify the most relevant tokens in attention, resulting in sub-linear time complexity with the context length while guaranting high end-to-end task accuracy. RetrievalAttention also reduces GPU memory consumption by offloading the index and KV vectors to CPU memory.

We thoroughly evaluate the accuracy and efficiency of RetrievalAttention across three long-context LLMs, using well-known long-context benchmarks like $\infty$-Bench [19] and RULER [20]. RetrievalAttention maintains the nearly same accuracy as full attention on different tasks and context lengths. Regarding the inference efficiency, for the 128K context on the RTX4090 GPU, RetrievalAttention achieves $7.93\times$ and $2.80\times$ decoding latency reduction compared to the retrieval method based on exact KNN and traditional ANNS index, respectively.

## 2 Background and Motivation

### 2.1 LLM and Long-Context Serving

Attention is the core component of the transformer-based LLM. In the generation process of the $t$-th token, the attention operation computes the dot product between the query vector $\mathbf{q}_t \in \mathbb{R}^{1\times d}$ (where $d$ is the hidden dimension) and the key vectors of all preceding tokens $\mathbf{k}_i \in \mathbb{R}^{1\times d}$ (for $i \leq t$). This product is scaled by $d^{-\frac{1}{2}}$ and normalized via a $\mathtt{Softmax}$ function to yield the attention weights $a_{t,i}$. Then it adds up values $\mathbf{v}_i$ with different weights, resulting in the output $\mathbf{o}_t$.

$$z_i = \frac{\mathbf{q}_t \cdot \mathbf{k}_i^T}{\sqrt{d}}, \quad a_{t,i} = \frac{e^{z_i}}{\sum_{j=1..t} e^{z_j}}, \quad \mathbf{o}_t = \sum_{i=1..t} a_{t,i} \cdot \mathbf{v}_i \tag{1}$$

LLM inference contains two stages: the prefill phase and decoding phase. The prefill phase, which only happens once, takes all tokens of the prompt as input and performs attention with a time-complexity $O(n^2)$. In the decoding (token generation) phase, the newly generated token is added to the input and computes attention scores with same complexity.

Due to the quadratic time complexity of attention computation, serving long-sequence input incurs extremely high costs. Table 1 shows the inference latency of Llama-3-8B without KV cache. When the prompt length reaches 1 million tokens, generating every token requires 1,765 seconds with over 96% of latency spent on attention operations. Since many long-context applications such as document QA and in-context learning involve a large proportion of fixed contexts, existing works [10, 9] pre-process these contexts offline by generating their KV cache and reusing them in successive prompts from users. The fixed contexts include documents, source codes, as well as system prompts or user prompts with domain-specific knowledge. Reusing the KV cache of these contexts across successive prompts effectively avoids redundant computation and reduce the time to first token.

However, the KV cache comes with a significant increase in GPU memory consumption and the attention complexity is still linear to the context length. As shown in Table 1, 125 GB memory is necessary for storing the KV cache when the context length reaches 1 million tokens, which is far beyond the GPU memory capacity of commodity GPUs such as the RTX4090 (24GB) or even high-end GPUs like A100 (40GB or 80GB). This necessitates either scaling to more GPUs to accommodate the large KV cache [21] or repetitively offloading and reloading the entire KV cache between CPU and GPU memory over PCIe [8], resulting in excessive communication overhead. Neither approach provides an efficient and cost-effective solution for long-context inference.

### 2.2 Dynamic and Sparse Attention

Corroborating recent work [13, 14], we observe that attention computation in LLMs exhibits significant sparsity. Despite the large context length, only a small fraction of tokens with the highest attention weights, also known as critical tokens, contribute significantly to the output. Figure 2a demonstrates the inherent sparsity, as measured by the mean squared error (MSE) of attention output between full attention and the sparse attention using only the top-$k$ critical tokens in the context of 128K tokens. The MSE drops sharply when $k$ is even smaller than 200 tokens, indicating the high sparsity in long-context processing.

Furthermore, we observe that as the model continues to generate new tokens, the positions of critical key vectors in the context dynamically change, heavily depending on the current query vector.

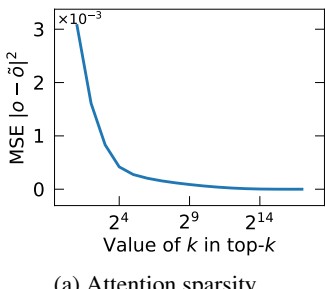

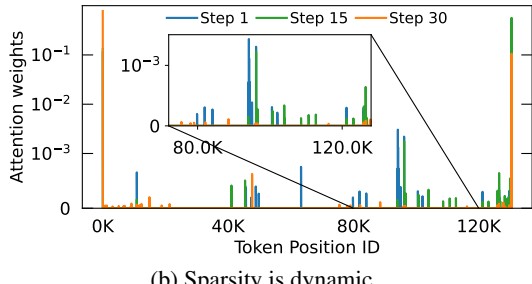

(a) Attention sparsity.                    (b) Sparsity is dynamic.

Figure 2: (a) Mean squared error (MSE) between full attention outputs and approximations based on the top-$k$ tokens in the 128K context. (b) Attention weights distribution across decoding steps.

Figure 2b illustrates the attention-weight distribution of the top-100 critical tokens across three decoding steps of a single inference process. The positions of those important tokens vary significantly, with only 30% overlap of these three steps. This highlights the dynamic and unpredictable nature of attention sparsity, necessitating an efficient and effective method for identifying critical tokens.

The dynamic sparsity shows a promising path to approximately compute attention with greatly reduced cost while maintaining model accuracy. For each query, if we can accurately identify the relevant key-value vectors with higher importance, minimum GPU memory and a much lower time complexity can be achieved for attention computation.

## 2.3 Challenges of Off-the-shelf Vector Search

Based on Equation 1, one key vector is critical for a query vector if they have a large inner product. With the inner product as a similarity function, performing searches on ANNS indexes aligns well with the goal of the attention mechanism to efficiently find critical key vectors to a query.

Traditional ANNS indexes generally cluster similar vectors and select the representative vector for each cluster [22] or directly build connections between similar vectors to form a proximity graph [23].[2] Both methods typically require scanning a limited subset of all vectors (e.g., 1%) to accurately identify the most similar vectors to the query, achieving high search efficiency and accuracy. Moreover, these indexes can be constructed offline based on fixed contexts (e.g., documents) and reused when serving different user queries. However, we find that naively applying off-the-shelf vector indexes fails to provide good performance because of the OOD issue between query ($Q$) and key vectors ($K$) in attention mechanism.

In conventional vector databases, the distribution of vectors between content and query is aligned (e.g., image search). Naively using traditional vector indexes for attention computation suffers from an inherent distribution gap between queries and keys, which are projected by different weights. Figure 3b (focus on $Q$ to $K$ for now) demonstrates the performance of widely-used vector indexes supported by Faiss [24] using a query vector to retrieve the most similar key vectors. It compares the percentage of keys scanned and the corresponding recall (i.e., the overlapping ratio between the retrieved top-100 results and the ground truth). Cluster-based IVF [22] requires scanning ∼30–50% data for a recall higher than 0.95, and graph-based HNSW [25] falls into a local optimum. The results highlight the challenge of performing efficient vector searches on attention data.

Fundamentally, the difficulty is due to the OOD between query and key vectors. To quantify this, we adopt the Mahalanobis distance [26], a standard metric widely used in vector search literature (e.g., OOD-DiskANN [27]) to measure the distance from a vector to a distribution. We sample 5,000 vectors from $Q$ and $K$ respectively as the query set and compute the Mahanobis distance from the vectors of $Q$ to the remaining vectors in $K$. Figure 3a shows that the vectors from $Q$ are significantly distant from the $K$ vectors (OOD) while $K$ themselves are very close. Therefore, traditional index building based solely on the closeness between key vectors does not align with the attention mechanism, which requires retrieving critical tokens as nearest neighbors from the query vectors' viewpoint. In contrast, Figure 3b shows that using sampled $K$ as the queries ($K$ to $K$) can easily achieve a high recall by only scanning 1–5% vectors because they are in the same distribution.

---

[2]In this context, we use "similar" and "close" to indicate vectors with larger inner product interchangeably.

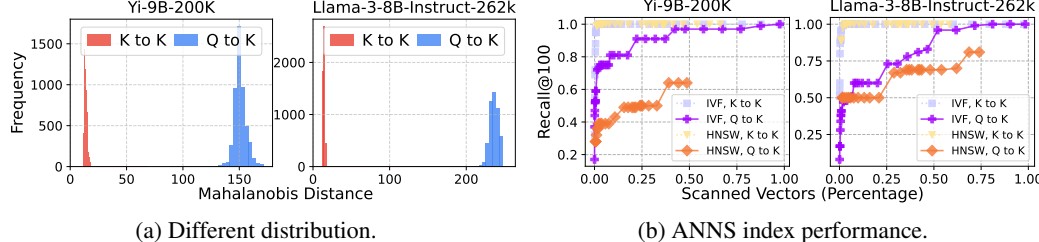

(a) Different distribution.
(b) ANNS index performance.

Figure 3: (a) Query vectors deviate from key vectors, while key vectors themselves are close. (b) Query vectors ($Q$) and key vectors ($K$) are dumped from Yi-9B and Llama-3-8B with a prompt length of 128,000 tokens. Off-the-shelf ANNS indexes perform poorly on $Q$ to $K$ searches while working well for $K$ to $K$ searches.

Similarly, query vectors in each attention head also follow the same distribution as they are projected by the same model weight. Therefore, for efficient vector search, the index must consider the OOD characteristic of the attention computation by design.

## 3 RetrievalAttention Design

In this work, we focus on accelerating the decoding phase of long contexts in scenarios where the fixed context takes a large proportion of the whole context (e.g., document QA). We propose RetrievalAttention that leverages attention-aware vector search to approximate attention computation by CPU-GPU co-execution. RetrievalAttention build vector indexes offline for the KV cache of fixed contexts (e.g., documents) and reuse them for successive prompts from users.

Figure 4a shows the overall design of RetrievalAttention. Based on our observation in Section 2.2, we derive an approximated attention by selectively retrieving relevant KV vectors while discarding those that are negligible (§3.1). To efficiently support long context, we store the pre-built vector indexes and KV vectors of fixed contexts in CPU memory, and use attention-aware vector search to find critical tokens during each decoding step (§3.2).

To better exploit the GPU devices, we leverage the attention pattern obtained in the offline index building phase to select a small subset of KV cache that is consistently important during the decoding phase and persist them (640 tokens of 128K context) on GPU devices. RetrievalAttention computes partial attention in parallel using dynamically retrieved KV vectors from CPU memory with persistent KV vectors stored in GPU memory and combines them together to get the final output (§3.3).

### 3.1 Approximated Attention

Based on the Equation 1, RetrievalAttention approximates the full attention output $\mathbf{o}_t$ by selectively utilizing the KV vectors associated with high attention weight (i.e., $a_{t,i}$). Specifically, we define $\mathcal{I}_{t,\epsilon}$ as a subset of token indices for which the attention weight surpasses $\epsilon$. Consequently, a sparse attention mechanism, which only considers tokens located in $\mathcal{I}_{t,\epsilon}$, can be defined as follows:

$$\mathbf{o}_t = \sum_{i \in \mathcal{I}_{t,\epsilon}} a_{t,i} \cdot \mathbf{v}_i + \cancel{\sum_{i \notin \mathcal{I}_{t,\epsilon}} a_{t,i} \cdot \mathbf{v}_i} \approx \sum_{i \in \mathcal{I}_{t,\epsilon}} \tilde{a}_{t,i} \cdot \mathbf{v}_i \quad \text{where} \quad \tilde{a}_{t,i} = \frac{e^{z_i}}{\sum_{j \in \mathcal{I}_{t,\epsilon}} e^{z_j}} \tag{2}$$

Based on the above approximation, we build RetrievalAttention to only consider important key-value vectors (i.e., $\mathcal{I}_{t,\epsilon}$) that are persistent in GPU cache and dynamically retrieved by vector indexes.

### 3.2 Attention-aware Vector Index

RetrievalAttention processes fixed contexts offline to generate the KV vectors and decides whether to build indexes for them (the decision method is elaborated in Section 3.3). All KV vectors with indexes are offloaded to CPU memory and used only when user queries comes.

To accelerate the vector search during decoding phase, RetrievalAttention diverges from traditional indexes that only consider the closeness between key vectors for index building. Instead, it leverages

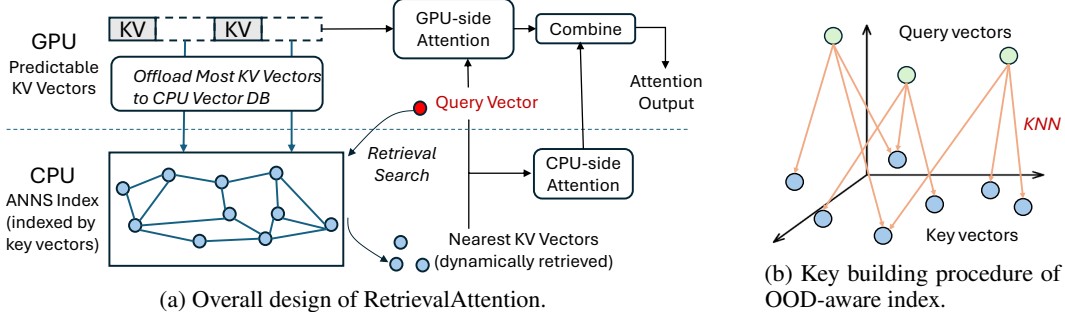

(a) Overall design of RetrievalAttention.

(b) Key building procedure of OOD-aware index.

Figure 4: (a) RetrievalAttention offloads most KV tokens to vector databases in CPU, which are retrieved during the decoding phase to find the most relevant KV tokens with queries. (b) During the index construction, we link each query to its exact top-$k$ nearest key vectors (KNN).

the existing query vectors to guide the index building for key vectors, efficiently mitigating the distribution gap. As shown in Figure 4b, during the index construction, RetrievalAttention explicitly establishes connections from the query vector to its nearest key vectors (i.e., exact $k$-nearest neighbors, or KNN), forming a mapping from query to key distribution. The KNN connections effectively serve as a bridge to reconcile their distribution differences. Using this structure, the decoding query vector can first search its nearest query vectors and then obtain the most relevant key vectors.

However, this structure still has imperfections in both memory overhead and search efficiency because we need to store and access query vectors besides key vectors. To address this problem, we leverage the projection technique from the state-of-the-art cross-modal ANNS index RoarGraph [28] to eliminate all query vectors. Specifically, we first project the KNN connections into key vectors by linking key vectors that are connected to the same query vectors, which efficiently streamlines the search. To enhance the connectivity of the graph structure, we then follow the conventional graph index building strategy to perform nearest neighbor search in the graph. If a key vector does not have enough neighbors, additional connections are added. The projection process connects key vectors that are perceived as close from the query vectors' perspective, allowing efficient index traversal for future decoding queries.

Our evaluation shows that, by effectively modeling the proximity relationship between the query and key vectors, the vector database only requires scanning 1–3% key vectors to reach high recall, significantly reducing the index search latency by 79% compared to the IVF indexes.

### 3.3 CPU-GPU Co-Execution

To exploit GPU parallelism and accelerate attention computation, RetrievalAttention decomposes the attention computation into two disjoint sets of KV cache vectors, the persistent ones on the GPU and the dynamic ones on the CPU, and combines the partial attention outputs during online inference.

We leverage the patterns observed in the offline phase to predict KV vectors that are consistently activated during token generation (i.e., static tokens). Similarly to StreamingLLM [13], our current implementation uses fixed initial tokens and the last sliding window of the context as the static pattern and persists them in the GPU cache. All remaining tokens require index building for retrieval.

During LLM inference, we first load KV vectors of the fixed context from CPU memory to GPU memory layer by layer to combine with the user query for context prefilling using full attention. When proceeding to the next layer, the KV cache of the previous layer, except for static tokens, is discarded to reduce GPU memory consumption. The time to first token is minimized because the attention computation for the fixed context is avoided by reusing their KV cache. The decoding phase leverages vector indexes for accelerating the token generation. In each decoding step, only a single query vector is sent to the CPU side for index traversal and attention computation by leveraging the compute capability of CPU, effectively reducing the data transfer over the PCIe. The partial attention output from CPU is returned to GPU and merged with the partial attention output on static tokens computed by GPU, inspired by the FlashAttention [29]. More detailed execution and algorithm description can be found in Appendix B.2.

# 4 Evaluation

In this section, we compare the performance of RetrievalAttention in long-context LLM inference against full attention and other state-of-the-art methods. Through experiments, we mainly explore the following questions: (1) **How does RetrievalAttention affect the model's inference accuracy?** Specifically, we assess the generation accuracy of RetrievalAttention and other methods across various downstream tasks (§4.2). (2) **Can RetrievalAttention efficiently reduce the token generation latency of long-context LLM inference?** We compare the end-to-end latency of RetrievalAttention with that of other baselines (§4.3).

## 4.1 Experimental Setup

**Testbed, Models, and Configurations.** We conduct experiments on a server equipped with one NVIDIA RTX4090 GPU (24GB memory) and an Intel i9-10900X CPU with 10 physical cores and 128GB DRAM. The experiment results using NVIDIA A100 GPU can be found in Appendix A.5. We implement RetrievalAttention on three long-context LLMs, including Llama-3-8B-Instruct-262k [30], Yi-9B-200K [31], and Yi-6B-200K [32].

**Baselines.** We compare RetrievalAttention with the following training-free baselines. (1) Full attention: it offloads the KV cache to CPU memory and selectively loads each layer's KV cache onto the GPU when needed. (2) StreamingLLM [13]: it retains initial tokens along with recent tokens and discards remaining tokens. (3) SnapKV [14]: it only caches the critical tokens observed from the last window of the prompt. (4) InfLLM [15]: it separates the KV cache into blocks and selects representative vectors for each block. The query scans all representative vectors and retrieves top-$k$ related blocks. (5) Quest [17]: it keeps track of the minimal and maximal values of KV cache pages and computes the criticality of a page using the query vector. (6) InfiniGen [33]: it prefetches the essential KV cache by speculating attention output for subsequent layers. (7) MagicPIG [34]: it uses local sensitive hashing to partition key vectors and hash collision for queries to get related keys. Detailed configurations of baselines can be found in Appendix A.1.

To better assess the effectiveness of our attention-aware index, we introduce two additional baselines using traditional vector search methods from Faiss [24]. Specifically, Flat is an exact KNN method that performs a linear scan of all key-value vectors, whereas IVF indexes key vectors through clustering. While HNSW is widely used in vector search, it always fails to retrieve the most relevant key vectors in attention scenarios across various parameter settings (Figure 3). Therefore, we exclude HNSW as a baseline due to its unsatisfactory end-to-end accuracy. By default, all indexing-based methods, including RetrievalAttention, retrieve the top-100 nearest key vectors and use 640 tokens (128 initial tokens + 512 local window tokens) on GPU as the static pattern. All evaluated methods are applied across all model layers for sparse attention during the decoding phase.

**Benchmarks.** We adopt three representative long-context benchmarks for evaluation. (1) RULER [20]: a comprehensive and widely used long-context benchmark consisting of retrieval, aggregation, QA tasks, and so on. (2) $\infty$-Bench [19]: this benchmark consists of retrieval tasks and realistic tasks. The average context length of $\infty$-Bench is over 100K tokens. (3) Needle-in-a-haystack [35]: it challenges the models to accurately retrieve information (the "needle") hidden within a lengthy document (the "haystack").

## 4.2 Accuracy on Long Context Tasks

**RULER.** Table 2 demonstrates that RetrievalAttention establishes state-of-the-art task accuracy across all evaluation scales, achieving nearly the same task accuracy as full attention in different context lengths. With the Llama-3-8B model, RetrievalAttention achieves 81.7% average accuracy (vs. full attention's 83.91%) with only a 2.21% performance drop. The loss of accuracy is only 0.52% on Yi-9B. Evaluations in Appendix A.2 also show similar superiority of RetrievalAttention on Yi-6B. In contrast, other training-free methods experience a noticeable reduction in accuracy, particularly for longer context sizes like 128K, as they fail to capture dynamically changed important tokens.

$\infty$**-Bench.** As shown in Tabel 3, RetrievalAttention achieves an average accuracy of $49.6\%$ across all tasks in $\infty$-Bench for Llama-3-8B, comparable to full attention ($50.4\%$). RetrievalAttention also achieves high accuracy on other models, demonstrating its improvement over existing token selection methods. While Flat and IVF sometimes achieve marginally higher accuracy, they incur

Table 2: Performance (%) of different methods and models on RULER.

| | Methods | Act. Tokens | Claimed | 16K | 32K | 64K | 128K | Avg. |
|---|---|---|---|---|---|---|---|---|
| Llama-3-8B | Full attention | 128K | 262K | 89.27 | 85.11 | 82.51 | 78.74 | 83.91 |
| | StreamingLLM | 2K | - | 20.99 | 16.36 | 12.52 | 11.34 | 15.30 (-68.61) |
| | SnapKV | 2K | - | 75.53 | 70.84 | 65.44 | 58.68 | 67.62 (-16.29) |
| | InfLLM | 2K | - | 38.29 | 32.44 | 27.94 | 25.71 | 31.10 (-52.81) |
| | InfiniGen | 2K | - | 87.16 | 81.60 | 72.52 | 64.10 | 76.35 (-7.56) |
| | Quest | 2K | - | 82.12 | 76.33 | 67.43 | 60.08 | 71.49 (-12.42) |
| | MagicPIG | 2K | - | 86.93 | 81.61 | 71.48 | 59.84 | 74.97 (-8.94) |
| | Flat | 640+100 | - | 87.01 | 84.97 | 80.99 | 74.34 | 81.83 (-2.08) |
| | IVF | 640+100 | - | 87.22 | 84.74 | 78.46 | 68.21 | 79.66 (-4.25) |
| | **RetrievalAttention** | 640+100 | - | 86.80 | 84.78 | 80.50 | 74.70 | **81.70(-2.21)** |
| Yi-9B | Full attention | 128K | 200K | 82.85 | 73.17 | 67.08 | 60.51 | 70.90 |
| | StreamingLLM | 2K | - | 19.08 | 13.48 | 12.53 | 12.81 | 14.48 (-56.42) |
| | SnapKV | 2K | - | 64.48 | 48.70 | 39.28 | 32.97 | 46.36 (-24.54) |
| | InfLLM | 2K | - | 36.17 | 28.20 | 22.65 | 20.94 | 26.99 (-43.91) |
| | InfiniGen | 2K | - | 82.45 | 72.87 | 62.10 | 56.76 | 68.55 (-2.35) |
| | Quest | 2K | - | 77.99 | 65.92 | 56.14 | 48.85 | 62.23 (-8.67) |
| | MagicPIG | 2K | - | 80.28 | 72.22 | 62.04 | 52.57 | 66.78 (-4.12) |
| | Flat | 640+100 | - | 84.42 | 74.58 | 66.16 | 59.50 | 71.17 (+0.27) |
| | IVF | 640+100 | - | 83.85 | 72.19 | 65.13 | 58.04 | 69.80 (-1.10) |
| | **RetrievalAttention** | 640+100 | - | 82.95 | 73.73 | 65.67 | 59.15 | **70.38 (-0.52)** |

significantly higher search costs (scanning 100% and 30% of all vectors, respectively) compared to RetrievalAttention (1-3%). Therefore, RetrievalAttention achieves competitive accuracy while drastically reducing search overhead, as reflected in the latency evaluation presented in Section 4.3. Detailed comparisons of different tasks are provided in Appendix A.3.

**Needle-in-a-haystack.** As shown in Figure 5, RetrievalAttention can effectively focus on information at various positions across different context windows, ranging from 4K to 128K. Evaluation under extremely (e.g., 1,000K) long-context shows that RetrievalAttention is capable of maintaining robust performance. The result is presented in Appendix A.4.

| Methods | Llama-3-8B | Yi-9B | Yi-6B |
|---|---|---|---|
| Full attention | 50.4 | 52.8 | 45.5 |
| StreamingLLM | 20.2 (-30.2) | 20.9 (-31.9) | 16.5 (-29.0) |
| SnapKV | 48.2 (-2.2) | 42.6 (-10.2) | 35.7 (-9.8) |
| InfLLM | 44.7 (-5.7) | 45.3 (-7.5) | 42.6 (-2.9) |
| InfiniGen | 47.5 (-2.9) | 48.8 (-4.0) | 44.9 (-0.6) |
| Quest | 47.6 (-2.8) | 47.9 (-4.9) | 44.5 (-1.0) |
| MagicPIG | 49.4 (-1.0) | 48.0 (-4.8) | 41.1 (-4.4) |
| Flat | 49.7 (-0.7) | 52.5 (-0.3) | 45.7 (+0.2) |
| IVF | 48.8 (-1.6) | 52.3 (-0.5) | 45.6 (+0.1) |
| **RetrievalAttention** | **49.6 (-0.8)** | **52.2(-0.6)** | **45.0 (-0.5)** |

Table 3: Performance (%) of different methods and models on ∞-Bench.

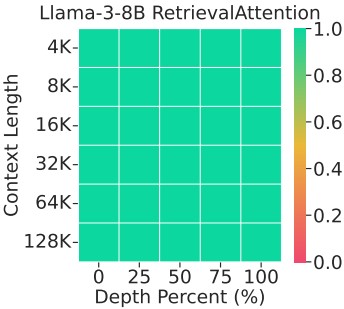

Figure 5: Performance of Llama-3-8B on Needle-in-a-haystack.

## 4.3 Inference Latency

In this section, we study the inference efficiency of RetrievalAttention using the combination of different fixed context lengths and user input lengths. Specifically, we first measure the decoding latency for an user input of 100 tokens while varying the length of the fixed context from 16K to 128K. Then we study the prefill and decoding latency with varying user input lengths (from 10 to 2K tokens), using a fixed context length of 128K tokens.

Figure 6 presents the decoding latency per token of RetrievalAttention and other baselines. As the context length increases, the decoding latency of full attention scales linearly, as loading the whole KV cache from CPU to GPU saturates the PCIe bandwidth. Because of fewer tokens involved in the attention computation, the latency of other baselines remains relatively low, but they suffer a

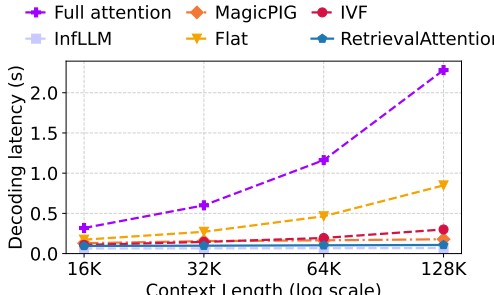

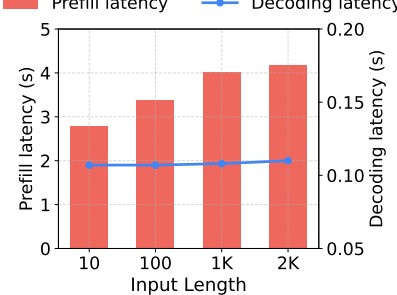

Figure 6: Decoding latency (s) as context length varies from 16K to 128K on Llama-3-8B.

Figure 7: Prefill and decoding latency (s) of varying input length on Llama-3-8B.

significant drop in model accuracy. Due to efficient attention-aware vector search, RetrievalAttention achieves $7.93\times$ and $2.80\times$ latency reduction compared to Flat and IVF for the 128K context, and the latency shows a slow growth with context length due to sub-linear time complexity of the index traversal. Benefiting from high recall with minimal data scanning, RetrievalAttention avoids memory bandwidth and computation contention caused by parallel retrieval across multiple heads on the CPU side. We present a detailed latency breakdown in Appendix A.6. Compared with Flat and IVF, RetrievalAttention effectively reduces the index search latency by 94% and 79%, respectively. As shown in Appendix A.5, this advantage becomes more pronounced with longer context lengths.

We compare both prefill and decoding latency as the user input ranges from 10 to 2K tokens under a fixed context of 128K tokens and show the results in Figure 7. During the prefill stage, RetrievalAttention loads the KV cache of the fixed contexts into the GPU memory to compute and generate the first token. The prefill latency increases slightly with the length of user input but remains low because the attention computation for the fixed context are avoided. Moreover, the decoding latency remains nearly constant, as short user inputs can be efficiently computed on the GPU together with the static pattern. Moreover, thanks to the parallel CPU-GPU design of RetrievalAttention, attention computation on the GPU could be effectively overlapped with CPU computation.

### 4.4 Sensitive Study and Micro Analysis

**Scalability**. Figure 8 presents the scalability of RetrievalAttention on Llama-3-8B model under 128K context length. The throughput improves by $1.6\times$ as the batch sizes increases. This improvement arises because RetrievalAttention offloads most of the KV cache to high-capacity CPU memory and transfers computation results of small size to the GPU. Full attention reaches a throughput of only 0.45 tokens per second, since the PCIe transfer of all the KV cache becomes a bottleneck even with a batch size of 1. The throughput of RetrievalAttention is eventually bound by the peak processing capacity of CPU.

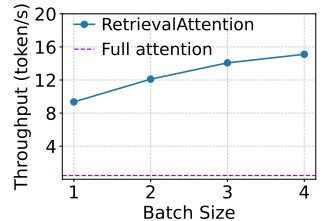

Figure 8: Throughput of different batch sizes on Llama-3-8B.

**Index building Time.** Table 4 presents the index building costs of RetrievalAttention for a context of 100K length. The costs mainly includes KNN search, projection and graph nodes linking. RetrievalAttention currently focus on scenarios with a largely fixed prefix and the index building is done offline, thereby not affecting prefill latency during online serving. We believe more advanced acceleration [36] deserves exploration to minimize the building cost for more dynamic use cases.

**Projection.** We study the effect of the projection technique by comparing RetrievalAttention's vector index with a vector index without projection. Specifically, the index without projection includes query vectors with connections to key vectors that identified as critical by query vectors. Searching on such graph structure requires routing between query and key vectors, resulting in excessive accesses. Table 5 presents the percentage of scanned vectors required by the two indexes for achieving a recall@100 of 0.95. The results clearly show that indexes without projection are inefficient.

Table 4: Index building time (s) of 100K context length.

| Model | Time (s) |
| --- | --- |
| Llama-3-8B | 788.2 |
| Yi-9B | 1121.2 |
| Yi-6B | 746.1 |

Table 5: The effect of projection in vector index quality.

| Model | Recall@100 | Scanned vectors |
| --- | --- | --- |
| | w/o projection / **ours** | w/o projection / **ours** |
| Llama-3-8B | 0.951 / **0.954** | 13.2% / **1.7%** |
| Yi-9B | 0.956 / **0.954** | 23.4% / **1.6%** |
| Yi-6B | 0.954 / **0.958** | 27.8% / **1.7%** |

**Static pattern size.** Table 6 presents RetrievalAttention's accuracy on RULER under variable static pattern sizes. The model we used is Llama-3-8B and the context length is 128K. As the size of static pattern increases, the accuracy gradually improves and tends to be stable. This is consistent with previous works that the number of sink tokens is relatively small [13]. The decoding latency remains unchanged as the static pattern is computed efficiently on the GPU. Moreover, our method also allows setting tokens at any position as a static pattern.

Table 6: Performance (%) of RetrievalAttention with different static pattern sizes on RULER.

| Sizes | Accuracy (%) |
| --- | --- |
| 16+64 | 73.93 |
| 32+128 | 74.15 |
| 64+256 | 74.52 |
| 128+512 | 74.70 |

## 5   Related Works

**Scaling Context Windows of LLMs.**   Recent studies have explored extending the context window of pre-trained LLMs, thereby enhancing their capability to handle more complex applications. Memorizing Transformer [37] and Focused Transformer [38] incorporate external memory mechanisms for context storage and leverage kNN-based retrieval to access relevant past information. However, they require additional model training to mitigate distraction issues. In contrast, our work builds upon existing long-context LLMs and enhances long-context inference efficiency without requiring any model retraining or architecture modification.

**KV Cache Compression.**   As the context length increases, the enormous KV cache not only puts a strain on GPU memory but also slows down inference speed, making it one of the most significant challenges in the long-context LLM inference. Some works [39, 40, 13, 41, 42, 14] attempt to prune the KV cache by leveraging the sparsity of attention. However, these methods often suffer from significant model accuracy drops due to the dynamic nature of attention sparsity. Some works [43, 44] quantize the KV cache size by approximating high-precision floating points with discrete low-bit values. These approaches are orthogonal to our work.

**Dynamic Sparse Attention.**   By identifying the dynamic nature of important KV vectors for different queries, recent works [17, 15, 33, 34, 45] retain all KV cache and dynamically attend to different parts of them. RetrievalAttention instead organizes the KV cache using attention-aware ANNS indexes, allowing the retrieval of important tokens with high recalls and low cost. Concurrent work PQCache [46] employs product quantization to compress KV vectors and retrieve critical tokens. Similarly, GPU-only SqueezedAttention [9] employs a K-means clustering strategy, similar to IVF. However, these methods fail to address the out-of-distribution (OOD) issue in attention mechanisms, often requiring retrieval of a large portion of the KV cache (e.g., 20% in PQCache) to maintain high model accuracy. Recent efforts [47, 48] further build on our observations to develop vector storage systems for end-to-end long-context acceleration.

## 6   Conclusion

We propose RetrievalAttention, a method that offloads KV vectors to CPU memory and leverages vector search for dynamic sparse attention to minimize inference cost. RetrievalAttention identifies the different distributions of the query and key vectors and employs an attention-aware approach to efficiently retrieve critical tokens for model generation. Experimental results demonstrate that RetrievalAttention effectively achieves $7.93\times$ and $2.80\times$ decoding speedup than exact KNN and traditional ANNS methods, on a single RTX4090 GPU for a context of 128K tokens. RetrievalAttention is the first system that supports long-context inference for 8B-level models on a single RTX4090 GPU (24GB) with acceptable latency and without compromising model accuracy.

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

# A Additional Experimental Details and Results

## A.1 Detailed Configurations of the Baselines

Detailed experimental configurations are as follows: (1) StreamingLLM: it uses 512 initial tokens and 1536 recent tokens for attention computation. (2) SnapKV: it employs an observation window of 32 tokens during the prefill phase and retains the top-2048 tokens. (3) InfLLM: it uses a block size of 128, 4 representative tokens per block, and 16 blocks in total. (4) Quest: it uses a page size of 64, 2 representative tokens per page, and 32 pages in total. (5) InfiniGen: it fetches the top-2048 tokens after speculative attention. (6) MagicPIG: it follows the original paper settings for $K$ and $L$ in different context lengths: 16K (7, 72), 32K (8, 95), 64K (9, 120), 128K (10, 150)

## A.2 Additional Results on RULER

Table 7 demonstrates the advantage of RetrievalAttention on Yi-6B, with only a 2.32% drop in accuracy. In contrast, other training-free methods suffer from larger accuracy degradation.

Table 7: Performance (%) of different methods on RULER of Yi-6B.

| | Methods | Act. Tokens | Claimed | 16K | 32K | 64K | 128K | Avg. |
|---|---|---|---|---|---|---|---|---|
| Yi-6B | Full attention | 128K | 200K | 69.12 | 61.64 | 58.36 | 55.77 | 61.22 |
| | StreamingLLM | 2K | - | 15.82 | 9.70 | 9.77 | 11.54 | 11.71 (-49.51) |
| | SnapKV | 2K | - | 45.36 | 36.11 | 33.43 | 29.53 | 36.11 (-25.11) |
| | InfLLM | 2K | - | 34.11 | 27.11 | 25.28 | 25.33 | 27.96 (-33.26) |
| | InfiniGen | 2K | - | 68.04 | 57.12 | 53.73 | 45.88 | 56.19 (-5.03) |
| | Quest | 2K | - | 63.18 | 56.05 | 48.32 | 43.20 | 52.69 (-8.53) |
| | MagicPIG | 2K | - | 66.61 | 57.69 | 51.10 | 44.04 | 54.86 (-6.36) |
| | Flat | 640+100 | - | 67.28 | 60.58 | 57.27 | 50.63 | 58.94 (-2.28) |
| | IVF | 640+100 | - | 67.00 | 58.94 | 55.99 | 50.31 | 58.06 (-3.16) |
| | **RetrievalAttention** | 640+100 | - | 67.49 | 59.46 | 57.20 | 51.44 | **58.90 (-2.32)** |

## A.3 Additional Results on ∞-Bench

As shown in Table 8, RetrievalAttention achieves comparable accuracy to the full attention, benefiting from its efficient dynamic retrieval of important tokens. Static methods, such as StreamingLLM and SnapKV, lack this capability and, therefore, achieve sub-optimal accuracy. During token generation phase, the critical tokens change dynamically according to the current query, invalidating the previously captured static patterns. InfiniGen exhibits a drop in model accuracy compared to full attention due to inaccurate speculation of important tokens from previous layers. Although InfLLM and Quest support dynamic retrieval of relevant blocks, they achieve nearly zero accuracy in complex tasks (e.g., KV retrieval) due to the low accuracy of representative vectors. MagicPIG achieves good task accuracy in ∞-Bench by employing LSH for critical token retrieval. However, the retrieval accuracy degrades in more challenging benchmarks such as RULER, as demonstrated in Table 2. Since RetrievalAttention can accurately identify the most relevant key vectors, it achieves the best accuracy in KV retrieval. Moreover, by retrieving more tokens (e.g., top-2000 shown in the column of Retr.KV) in KV retrieval, RetrievalAttention achieves nearly the same accuracy as full attention, which demonstrates the effectiveness of our method in complex and dynamic tasks.

It is worth noting that Flat and IVF need to scan 100% and 30% of the key vectors to achieve the same task accuracy as RetrievalAttention. In contrast, RetrievalAttention only requires scan 1–3% vectors, resulting in much lower decoding latency.

## A.4 Performance of Extremely Long-context Inference

Figure 9 shows the evaluation results of RetrievalAttention for extremely long contexts using the Llama-3-8B-1048K model. RetrievalAttention still passes all test cases when ranging the context length from 250K to 1,000K, which demonstrates the robustness of our attention-aware indexes.

Table 8: Performance (%) of different methods and models on ∞-Bench. The size of the static pattern is consistently 640 (128 initial tokens + 512 tokens in the local window). All indexing-based methods, including Flat, IVF, and RetrievalAttention retrieve top-100 key vectors by default. In the relatively complicated task KV Retrieval, we include the results of retrieving top-2000 key vectors.

| | Methods | Act. Tokens | Retr.N | Retr.P | Retr.KV | Code.D | Math.F | En.QA | En.MC | Avg. |
|---|---|---|---|---|---|---|---|---|---|---|
| **Llama-3-8B** | Full attention | 128K | 100.0 | 100.0 | 17.5 | 19.0 | 39.5 | 9.1 | 68.0 | 50.4 |
| | StreamingLLM | 2K | 5.0 | 5.0 | 1.0 | 18.5 | 40.0 | 6.0 | 66.0 | 20.2 (-30.2) |
| | SnapKV | 2K | 100.0 | 100.0 | 0.5 | 18.0 | 40.0 | 11.8 | 67.0 | 48.2 (-2.2) |
| | InfLLM | 2K | 100.0 | 100.0 | 0.5 | 20.5 | 48.0 | 7.0 | 37.0 | 44.7 (-5.7) |
| | InfiniGen | 2K | 100.0 | 100.0 | 0.5 | 19.0 | 40.5 | 8.1 | 64.5 | 47.5 (-2.9) |
| | Quest | 2K | 100.0 | 100.0 | 0.0 | 18.0 | 40.0 | 8.2 | 67.0 | 47.6 (-2.8) |
| | MagicPIG | 2K | 99.5 | 100.0 | 9.5 | 19.5 | 41.0 | 8.7 | 67.5 | 49.4 (-1.0) |
| | Flat | 640+100/2K | 100.0 | 100.0 | 8.5/14.5 | 19.0 | 40.0 | 7.5 | 67.0 | 48.9 (-1.5) / 49.7 (-0.7) |
| | IVF | 640+100/2K | 94.0 | 100.0 | 9.5/14.0 | 19.0 | 40.0 | 7.8 | 67.0 | 48.2 (-2.2) / 48.8 (-1.6) |
| | **RetrievalAttention** | 640+100/2K | 100.0 | 100.0 | 9.0/14.0 | 19.0 | 40.0 | 7.5 | 67.0 | **48.9 (-1.5) / 49.6 (-0.8)** |
| **Yi-9B** | Full attention | 128K | 100.0 | 100.0 | 30.5 | 25.5 | 36.5 | 9.8 | 67.0 | 52.8 |
| | StreamingLLM | 2K | 5.0 | 5.0 | 0.5 | 24.0 | 33.5 | 6.4 | 72.0 | 20.9 (-31.9) |
| | SnapKV | 2K | 63.0 | 100.0 | 0.5 | 23.0 | 33.0 | 10.3 | 68.5 | 42.6 (-10.2) |
| | InfLLM | 2K | 100.0 | 100.0 | 0.5 | 20.5 | 43.0 | 9.4 | 44.0 | 45.3 (-7.5) |
| | InfiniGen | 2K | 100.0 | 98.5 | 8.0 | 24.0 | 35.5 | 10.3 | 65.5 | 48.8 (-4.0) |
| | Quest | 2K | 99.5 | 100.0 | 0.0 | 24.0 | 32.0 | 11.0 | 69.0 | 47.9 (-4.9) |
| | MagicPIG | 2K | 75.0 | 99.0 | 18.5 | 28.5 | 37.0 | 10.9 | 66.5 | 48.0 (-4.8) |
| | Flat | 640+100/2K | 100.0 | 100.0 | 21.0/30.0 | 23.0 | 35.0 | 10.8 | 68.5 | 51.2 (-1.6) / 52.5 (-0.3) |
| | IVF | 640+100/2K | 99.0 | 100.0 | 19.5/29.5 | 23.0 | 35.0 | 10.7 | 69.0 | 50.9 (-1.9) / 52.3 (-0.5) |
| | **RetrievalAttention** | 640+100/2K | 99.5 | 100.0 | 20.0/30.0 | 23.0 | 35.0 | 9.5 | 68.5 | **50.8 (-2.0) / 52.2 (-0.6)** |
| **Yi-6B** | Full attention | 128K | 98.0 | 100.0 | 3.5 | 31.0 | 11.0 | 19.2 | 55.5 | 45.5 |
| | StreamingLLM | 2K | 5.0 | 5.0 | 0.5 | 27.5 | 11.0 | 12.2 | 54.0 | 16.5 (-29.0) |
| | SnapKV | 2K | 39.0 | 100.0 | 0.0 | 30.5 | 8.5 | 17.1 | 55.0 | 35.7 (-9.8) |
| | InfLLM | 2K | 99.0 | 100.0 | 0.5 | 27.5 | 18.0 | 12.7 | 40.5 | 42.6 (-2.9) |
| | InfiniGen | 2K | 99.0 | 100.0 | 1.5 | 30.0 | 11.5 | 18.1 | 54.0 | 44.9 (-0.6) |
| | Quest | 2K | 98.5 | 100.0 | 0.0 | 31.0 | 10.0 | 17.8 | 54.5 | 44.5 (-1.0) |
| | MagicPIG | 2K | 69.5 | 99.5 | 1.5 | 31.5 | 12.0 | 19.3 | 54.5 | 41.1 (-4.3) |
| | Flat | 640+100/2K | 98.5 | 100.0 | 2.5/3.0 | 30.5 | 16.0 | 17.7 | 54.5 | 45.7 (+0.2) / 45.7 (+0.2) |
| | IVF | 640+100/2K | 98.0 | 100.0 | 2.5/3.5 | 29.5 | 16.0 | 17.5 | 54.5 | 45.4 (-0.1) / 45.6 (+0.1) |
| | **RetrievalAttention** | 640+100/2K | 95.0 | 99.0 | 3.0/3.0 | 30.0 | 16.0 | 17.6 | 54.5 | **45.0 (-0.5) / 45.0 (-0.5)** |

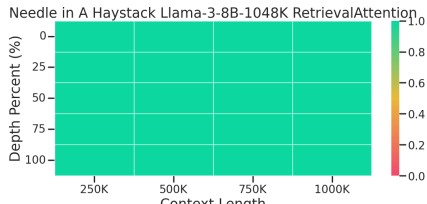

Figure 9: Performance of RetrievalAttention on Needle-in-a-haystack test of 1,000K context.

## A.5 Decoding Latency on A100 GPU

We test the generality of RetrievalAttention by measuring its performance on a server with one A100 GPU (40GB) and one AMD EPYC 7V13 CPU with 12 physical cores and 220GB DRAM. We show the token generation latency of different methods on three models in Figure 10. Full attention with KV cache offloading suffers from the PCIe transmission bottleneck. InfLLM achieve faster decoding speed, but this is at the expense of a significant drop in model accuracy. In contrast, RetrievalAttention does not compromise generation accuracy while achieving much lower decoding latency than Flat and IVF because of the efficient mitigation of the OOD problem.

We also evaluate how the decoding latency changes when the context length varies from 100K to 1M tokens on Llama-3-8B model and the results can be found in Figure 11. To make sure there is enough CPU memory to hold the KV cache and indexes, especially in the 1M context scenario, we use a powerful machine equipped with an AMD EPYC 7V12 CPU with 48 cores and 1.72 TB of memory. The machine is also equipped with the same 40GB A100 GPU. Different from Flat, IVF, and MagicPIG, whose latency numbers are sensitive to context size, RetrievalAttention only has a minor latency increase when the context size increases 10× from 100K to 1M.

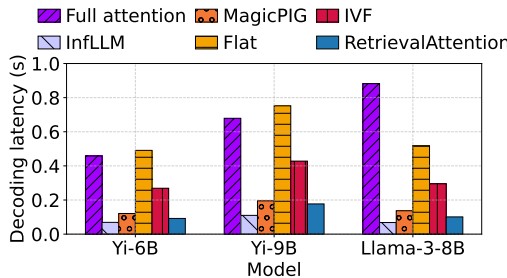
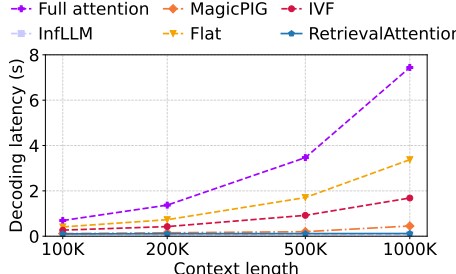

Figure 10: Decoding latency (s) of 128K context length on A100 GPU.

Figure 11: Decoding latency (s) of Llama-3-8B as context length varies from 100K to 1M.

## A.6 Detailed Latency Breakdown

Table 9 and Table 10 show a detailed breakdown of decoding latency on the RTX4090 GPU. Thanks to its decoupled CPU-GPU co-execution architecture, RetrievalAttention performs partial attention computation on both the CPU and GPU, and transfers the CPU-side results to the GPU for final combination. The data transfer volume is extremely low. As shown in Table 10, the PCIe transfer overhead accounts for only 1.8% of the end-to-end latency with a 128K context. Moreover, the computations of CPU and GPU could be fully overlapped.

Table 9: Decoding latency breakdown (s) on Llama-3-8B.

| Methods | Retrieval | CPU Attn | Others | Total |
|---|---|---|---|---|
| Flat | 0.791 | 0.016 | 0.041 | 0.848 |
| IVF | 0.243 | 0.017 | 0.040 | 0.300 |
| **RetrievalAttention** | 0.051 | 0.016 | 0.040 | 0.107 |

Table 10: Detailed breakdown (s) of each component.

| | Retrieval | CPU Attn | GPU Attn | PCIe Comm. | Result Combination | Other Layers | Total |
|---|---|---|---|---|---|---|---|
| Latency (s) | 0.051 | 0.016 | 0.006 | 0.002 | 0.006 | 0.026 | 0.107 |

## A.7 Scalability with CPU Cores

Figure 12 demonstrates the scalability of RetrievalAttention. The model we used is Llama-3-8B and the context length is 128K. The decoding latency decreases with more CPU cores. This stems from the ability to perform concurrent vector searches across attention heads.

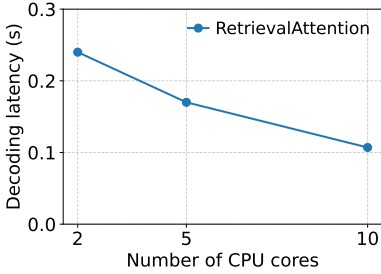

Figure 12: Decoding latency (s) under different CPU cores.

## A.8 Index Recall vs. Scanning Vectors

We analyze the efficiency of attention-aware vector index by examining the relationship between recall and the number of scanned key vectors. The number of key vectors scanned to achieve a target

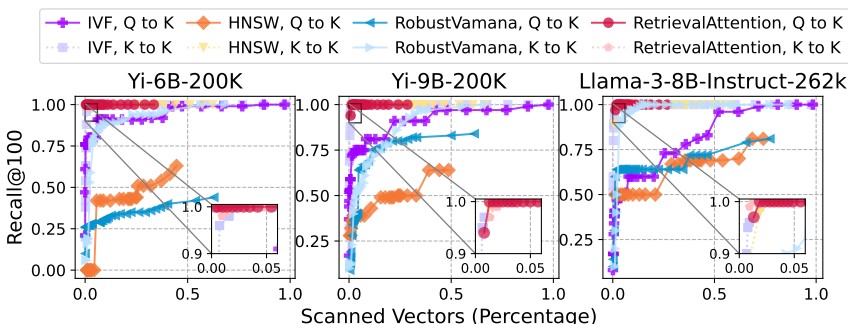

Figure 13: Recall vs. scanning key vectors when using $Q$ or $K$ as the query.

recall serves as an indicator of search efficiency (upper left is better). Figure 13 demonstrates that for the $Q$ to $K$ search ($qK^T$ in attention), RetrievalAttention requires scanning only a very limited number of key vectors (1–3%) to reach recall@100 higher than 0.95, whereas traditional indexes necessitate retrieving a significantly higher number of keys. We also included an OOD-optimized solution, RobustVamana [49], for comparison. However, it performs poorly on attention vectors. The efficiency of RetrievalAttention arises because it effectively mitigates the OOD issue between query and key vectors in attention. In contrast, for the in-distribution $K$ to $K$ search, all indexes exhibit good performance.

## A.9 Dynamic Retrieval Budget Allocation

We investigate the impact of adjusting the retrieval budget based on the sparsity degree across layers, using the budget allocation policy from PyramidKV [50]. We compare the performance of RetrievalAttention with and without the PyramidKV-based budget allocation strategy on the $\infty$-Bench benchmark. For original RetrievalAttention, we set a fixed budget of 2000 tokens for all heads in all layers. In contrast, PyramidKV dynamically adjusts the retrieval size across different layers, allocating more in lower layers and less in higher ones.

The results in Table 11 show that PyramidKV allocation strategy achieves better performance in Retr.KV tasks, though it slightly decreases performance in the En.QA task. On average, the accuracy of dynamic budget allocation strategy slightly surpasses the original RetrievalAttention on Llama-3-8B. This indicates that dynamic budget allocation is promising, though it may require adjustment based on specific tasks.

Table 11: Performance (%) of RetrievalAttention and RetrievalAttention w/ PyramidKV on $\infty$-Bench.

| Methods | Retr.N | Retr.P | Retr.KV | Code.D | Math.F | En.QA | En.MC | Avg. |
|---|---|---|---|---|---|---|---|---|
| Full Attention | 100.0 | 100.0 | 17.5 | 19.0 | 39.5 | 9.1 | 68.0 | 50.4 |
| **Ours** | 100.0 | 100.0 | 14.0 | 18.5 | 40.0 | 8.7 | 67.5 | 49.8 |
| **Ours w/ PyramidKV** | 100.0 | 100.0 | 16.0 | 18.5 | 40.0 | 8.5 | 67.5 | 50.1 |

## A.10 Performance on the Larger Models

To demonstrate the generalizability of RetrievalAttention on larger models, we evaluate it on the Llama-3-70B-262k using a server equipped with eight 40GB A100 GPUs. We choose the most complex task, KV retrieval in $\infty$-Bench, to stress test the efficiency of RetrievalAttention.

The results in Table 12 show that RetrievalAttention achieves nearly the same task accuracy as the exact KNN method, Flat, and outperforms Quest. The decoding speed of RetrievalAttention is faster than Flat as it effectively reduces the number of vectors to be scanned.

## A.11 Index Quality

The index quality and construction time is a trade-off by adjusting parameters such as graph node degree and node linking traversal depth during index building. We conduct an experiment of "the

Table 12: Performance (%) and decoding latency (s) of different methods with Llama-3-70B on the complex task KV retrieval in ∞-Bench.

|  | Full attention | StreamingLLM | Quest | Flat | **RetrievalAttention** |
|---|---|---|---|---|---|
| Accuracy (%) | 35.0 | 0.0 | 13.0 | 24.0 | 23.5 |
| Decoding latency (s) | 14.8 | 0.14 | 1.36 | 5.48 | 1.52 |

number of accessed tokens to achieve recall@100=1.0 during search" versus "construction time" presented in Table 13. All numbers are scaled based on the index used in our paper (i.e., the construction time and the number of accessed tokens for the index used in our paper are represented as 1). We observe that when index construction accounts for 51% of the total time, search efficiency drops by $2.21\times$, as more tokens are involved in the computation. This reflects a trade-off between search efficiency and index construction overhead in the current implementation.

Table 13: Trade-off between construction time and required scanned tokens (normalized results).

| Construction Time | 1.00 | 0.84 | 0.73 | 0.63 | 0.51 | 0.48 | 0.45 |
|---|---|---|---|---|---|---|---|
| Scanned Tokens | 1.00 | 1.52 | 1.72 | 2.09 | 2.21 | 3.50 | 5.17 |

# B  RetrievalAttention Algorithm

## B.1  Formula of Combining Attention Results from the CPU and GPU Side

RetrievalAttention partitions the KV vectors for attention into two disjoint sets: persistent ones on GPU (denoted as $\mathcal{W}$) and dynamically retrieved ones on CPU (denoted as $\Omega$).

$$\mathcal{I}_{t,\epsilon} = \mathcal{W} \cup \Omega \tag{3}$$

Attention operation is applied to the two sets of KV vectors separately on CPU and GPU, generating two partial attention outputs (denoted as $\mathbf{o}_{\mathcal{W}}$ and $\mathbf{o}_{\Omega}$, respectively). To guarantee the approximated attention output equals to the attention computation on $\mathcal{I}_{t,\epsilon}$, RetrievalAttention uses a similar idea of FlashAttention [29] to combine $\mathbf{o}_{\mathcal{W}}$ and $\mathbf{o}_{\Omega}$ in the following equations:

$$\mathbf{o}_{\mathcal{W}} = \text{Attn}(\mathbf{q}_t, \mathbf{K}[\mathcal{W}, :], \mathbf{V}[\mathcal{W}, :])$$
$$= \frac{\sum_{i \in \mathcal{W}} e^{z_i - \tilde{z}_1} \cdot v_i}{\sum_{i \in \mathcal{W}} e^{z_i - \tilde{z}_1}}$$
$$\mathbf{o}_{\Omega} = \text{Attn}(\mathbf{q}_t, \mathbf{K}[\Omega, :], \mathbf{V}[\Omega, :])$$
$$= \frac{\sum_{i \in \Omega} e^{z_i - \tilde{z}_2} \cdot v_i}{\sum_{i \in \Omega} e^{z_i - \tilde{z}_2}}$$
$$\mathbf{o}_t = \gamma_1 \cdot \mathbf{o}_{\mathcal{W}} + \gamma_2 \cdot \mathbf{o}_{\Omega} \tag{4}$$

where $\tilde{z}_1 = \max_{i \in \mathcal{W}} z_i$ and $\tilde{z}_2 = \max_{i \in \Omega} z_i$ are the local maximum dot products in set $\mathcal{W}$ and $\Omega$ respectively. And $\gamma_1$ and $\gamma_2$ are re-scaling factors to guarantee the attention output is the same as that on $\mathcal{I}_{t,\epsilon}$, which are defined as follows:

$$\gamma_1 = \frac{e^{\tilde{z}_1 - \tilde{z}} \cdot \sum_{i \in \mathcal{W}} e^{z_i - \tilde{z}_1}}{\sum_{i \in \mathcal{I}_{t,\epsilon}} e^{z_i - \tilde{z}}}$$
$$\gamma_2 = \frac{e^{\tilde{z}_2 - \tilde{z}} \cdot \sum_{i \in \Omega} e^{z_i - \tilde{z}_2}}{\sum_{i \in \mathcal{I}_{t,\epsilon}} e^{z_i - \tilde{z}}} \tag{5}$$

## B.2  Overall Execution Flow

Algorithm 1 summarizes the design of RetrievalAttention and elaborates the procedure in an algorithm. (1) Offline vector index construction. RetrievalAttention pre-computes the KV vectors and builds the

attention-aware vector index offline for fixed contexts. The index and KV vectors are stored in CPU memory once the construction is completed. (2) Prefill phase. RetrievalAttention loads the cached KV vectors from CPU to GPU and computes full attention for the relatively short new user queries (#6). Since the number of these new tokens is relatively small, we store the corresponding KV cache on the GPU with static patterns (#7). (3) Decoding phase. RetrievalAttention performs vector search over the index for each query vector on the CPU, transfers the results to the GPU, and combines them with the GPU-computed results to obtain the final output. In detail, RetrievalAttention computes partial attention using the FlashAttention [29] kernel (#8) for static pattern KV vectors that are always stored in the GPU. In parallel with GPU computation, RetrievalAttention leverages the specially designed attention-aware vector index to find the most relevant KV vectors and computes attention on CPU (#9 - #10). Finally, RetrievalAttention combines the partial attention outputs on GPU and CPU using Equation (4) and gets the approximated attention output (#11).

---

**Algorithm 1:** RetrievalAttention

---

**Input:** Query vector $\mathbf{q}_t \in \mathcal{R}^{1 \times d}$
**Data:** KV Cache in GPU $\mathbf{K}_{\mathcal{W}}, \mathbf{V}_{\mathcal{W}} \in \mathcal{R}^{|\mathcal{W}| \times d}$
**Data:** CPU-based Vector Database $\mathcal{H}$
**Output:** Attention output $\mathbf{o}_t \in \mathcal{R}^{1 \times d}$
```
// Offline: Find the predictable KV vectors and
   build index
```
1   $\mathcal{W}' \leftarrow \text{PredictStaticActiveTokens}(...)$;
2   **for** $\{i | i \in \mathcal{H} \cup \mathcal{W}'\}$ **do**
    `// Will be computed on GPU`
3      $\mathcal{H}.\text{remove}(i)$;   $\mathcal{W}.\text{insert}(i)$;

4   **for** $\{i | i \notin \mathcal{W}' \wedge i \in \mathcal{H}\}$ **do**
    `// Will be retrieved from CPU`
5      $\mathcal{W}.\text{remove}(i)$;   $\mathcal{H}.\text{insert}(i)$;

```
// Online prefill user request
```
6   $U_K, U_V \leftarrow \text{PrefillUserRequest}(...)$;
7   $\mathcal{W}.\text{insert}(k, v | k \in U_K, v \in U_V)$;
```
// Online Decoding
// Attention on GPU
```
8   $\mathbf{o}_{\mathcal{W}} \leftarrow \text{FlashAttention}(\mathbf{q}_t, \mathbf{K}_{\mathcal{W}}, \mathbf{V}_{\mathcal{W}})$
```
// Search in vector database to retrieve most
   relevant KV vectors to compute attention on CPU
```
9   $\Omega \leftarrow \text{VectorSearch}(\mathbf{q}_t)$;
10   $o_{\Omega} \leftarrow \text{AttentionCPU}(\Omega)$;
```
// Combine partial attention outputs
```
11   $\mathbf{o}_t = \gamma_1 \cdot \mathbf{o}_{\mathcal{W}} + \gamma_2 \cdot \mathbf{o}_{\Omega}$; `// Equation 4,5`

---

## C   Theoretical Proofs of Retrieval Accuracy or Complexity Bounds

Existing widely-used vector retrieval methods, including graph and cluster-based indexes, typically do not guarantee retrieval accuracy because their designs are heuristic-based on vector proximity. We follow common practice by stress-testing retrieval accuracy through extensive empirical evaluations. The results indicate that RetrievalAttention provides better and more robust retrieval accuracy than other methods.

**Index building time complexity:** (1) KNN search for matching query vectors to key vectors takes $O(n^2 \cdot d)$ for the context length of $n$ and the vector dimensionality of $d$. (2) Projection node linking on CPU have a complexity of $O(n \cdot C) + O(n \cdot \log n)$, where the constant $C$ is related to the $K$ in KNN and node linking requires traversing all $n$ nodes. The $O(n \cdot \log n)$ part refers to a connectivity enhancement following traditional graph-based indexing methods.

**Index search time complexity:** Assume the projection technique addresses the OOD problem (empirically verified), the time complexity of the index search is close to $O(\log n)$, similar to other graph-based methods.

## D Additional Related Work

**Sparse Transformers.** Since the quadratic complexity of attention has become the bottleneck of LLM efficiency for long context applications, numerous works have been studied to design sparse transformers to reduce the computational and memory complexity of the self-attention mechanism. Some works restrict the attention computation to predefined patterns, including sliding windows [51], dilated windows [52], or a mixture of different patterns [53, 54, 55]. Some approaches use cluster-based sparsity based on hash value [56] or KNN algorithms [57, 58]. These solutions either require pre-training a model from scratch or target limited scenarios like CPU-only, which do not work for our target to out-of-box usage of LLMs on the GPU-CPU architecture. Although some approaches [15, 16] exploit the dynamic sparse nature of LLMs, they often use some estimation using low-rank hidden states or post-statistical approaches, which incurs high overhead but with low accuracy. Moreover, all these approaches have to maintain full KV vectors on GPU with only accelerated inference by reduced memory movement, which does not solve the challenge of limited GPU memory.

## E Broader Impacts

RetrievalAttention bridges the fields of efficient attention and vector retrieval. It significantly accelerates the decoding speed and reduces GPU memory consumption for long-context LLM serving, facilitating their deployment and practical application. By reducing latency and memory usage, RetrievalAttention lowers the costs of long context serving, and thus contributes to the democratization of advanced AI. Moreover, it encourages further research and innovation in related areas.

## F Limitations

RetrievalAttention is designed to accelerate decoding and reduce GPU memory consumption in long-context inference. It focuses on scenarios where fixed contexts are reused across multiple generations, allowing offline index construction. The prefill phase is not optimized in the current design. Exploring efficient index construction strategies to support more general long-context inference remains an important direction for future work.

