# OpenReview forum: "RetrievalAttention: Accelerating Long-Context LLM Inference via Vector Retrieval"
_NeurIPS.cc/2025/Conference — NeurIPS 2025 poster_

### Official Review · Reviewer_krY7 · 2025-06-30

**Clarity:** 4
**Significance:** 4
**Originality:** 4
**Rating:** 5
**Confidence:** 4

**Summary:**

This paper presents RetrivalAttention, which dynamically retrieves the KV cache to the GPU during inference to reduce the GPU memory usage and speed up the inference.

Unlike traditional methods that do static or dynamic KV cache eviction, this paper offloads the KV cache to the CPU and builds a vector-search index that helps nearest neighbor search and retrieval during the decoding process.

Evaluation results show that the proposed solution can help run very long context inference on a consumer-grade GPU with limited memory, without causing too much quality drop.

**Questions:**

1. What's the detailed setup of "full attention" in Figure 6? Why does the decoding latency grow superlinearly as the context length increases? Shouldn't the time complexity be O(N)?

2. I'm wondering if the proposed solution can increase the maximum batch size during the LLM inference, and thus improve the overall LLM inference throughput?

**Ethical Concerns:**

["NO or VERY MINOR ethics concerns only"]

**Final Justification:**

I think this is a good paper and should qualify for NeurIPS, so the final score is 5

**Quality:**

3

**Strengths And Weaknesses:**

# Strengths

1. It's innovative to use the vector-search methods for the KV cache retrieval during the LLM inference.
2. The evaluation is solid, including both quality and system efficiency evaluation.

# Weaknesses

1. Clarity issue about the evaluation setup: the "full attention" baseline is not clearly explained in the evaluation setup.

---

> ### Author Rebuttal · Authors · 2025-07-31
>
> We sincerely appreciate your constructive feedback, which is invaluable for further improving our paper. We address each review comment below.
>
> W1.***"Clarity issue about the evaluation setup: the "full attention" baseline is not clearly explained in the evaluation setup."***
>
> Q1.***"What's the detailed setup of "full attention" in Figure 6? Why does the decoding latency grow superlinearly as the context length increases? Shouldn't the time complexity be O(N)?"***
>
>
> We apologize for the confusion of the "full attention" baseline. We will clarify this in the revised version of the paper.
>
> For the long-context inference on memory-constrained GPU devices, there are two common strategies for "full attention" implementation while avoiding Out-of-Memory (OOM) errors: KV recomputation or KV cache offloading to CPU memory. For the KV recomputation, the KV vectors are recomputed for all previous tokens at each decoding step, leading to quadratic computation complexity and high decoding latency. For example, decoding a single token with a 128K context length takes 43 seconds. This makes it impractical for real-time inference in long-context scenarios.
>
> To mitigate the high latency of recomputation, we implemented "full attention" using KV cache offloading. Specifically, we offload the entire KV cache to the CPU memory and selectively load the KV cache for one model layer onto the GPU on demand during inference. This strategy drastically reduces the decoding latency to approximately 2.3 seconds per token for a 128K context, making it a more feasible baseline for comparison in long-context settings.
>
> Regarding the observed superlinear growth in decoding latency for "full attention" in Figure 6, we would like to clarify that the x-axis (Context Length) in the figure is plotted on a log scale for better clarity across a wide range of context lengths. While the plot visually suggests a non-linear trend of full attention, the actual decoding latency of "full attention" has an O(N) time complexity with respect to the context length N. This is because the KV cache size grows linearly with the context length, and consequently, both the data transfer latency over PCIe and the attention computation also scale linearly with N.
>
> To address this, we will explicitly state in the caption of Figure 6 that the x-axis uses a log scale. We are also open to re-plotting the figure with a linear x-axis.
>
>
> Q2.***"I'm wondering if the proposed solution can increase the maximum batch size during the LLM inference, and thus improve the overall LLM inference throughput?"***
>
> Thank you for this insightful question. Yes, our proposed solution indeed enables a larger maximum batch size during LLM inference, thereby improving overall LLM inference throughput. This is because RetrievalAttention can offload most KV cache to the CPU memory with high capacity.
>
> We conducted experiments to demonstrate this capability, and results are presented in Appendix A.9 of our original submission. It shows that as the batch size increases, the decoding throughput of our method increases by approximately 1.6$\times$. The throughput eventually becomes constrained by the CPU's compute power.
>
> We will highlight this more prominently in the revised manuscript.

---

### Official Review · Reviewer_6mEN · 2025-06-30

**Clarity:** 3
**Significance:** 3
**Originality:** 2
**Rating:** 4
**Confidence:** 4

**Summary:**

This paper presents RetrievalAttention, a system that accelerates long-context inference by offloading the KV cache to a CPU-based vector index. The work provides a strong analysis showing that standard ANNS methods fail due to an OOD problem between query and key vectors. The core contribution is an attention-aware index that adapts a projection technique from prior work to solve this issue, achieving significant speedups with minimal accuracy loss.

**Questions:**

- why is a simple IVF often comparable to RetrievalAttention? (e.g. Table 3, Yi-*B)

**Ethical Concerns:**

["NO or VERY MINOR ethics concerns only"]

**Final Justification:**

authors address all the concerns so keeping the rating

**Limitations:**

yes

**Quality:**

3

**Strengths And Weaknesses:**

**S1.** Provides a simple and effective system design for efficient long-context inference.

**S2.** The paper is easy to follow, with a clear and compelling motivating analysis of the OOD problem.

**W1.** The paper's core technique is heavily adapted from RoarGraph [28] but isn't explained in detail, making the work not self-contained and its novelty appear limited.

**W2.** Key design choices, such as the vector projection technique and the static token heuristic, lack comparative ablation studies to justify their selection over other alternatives.

---

> ### Author Rebuttal · Authors · 2025-07-31
>
> We sincerely appreciate the reviewer's constructive feedback. As recognized by the reviewer, our key contribution lies in the novel application of attention-aware vector indexes to long-context LLM inference, coupled with systems designs for both high accuracy and efficiency. We will take the review comments to further improve the paper. We address each review comment below.
>
> W1. ***"The paper's core technique is heavily adapted from RoarGraph [28] but isn't explained in detail, making the work not self-contained and its novelty appear limited."***
>
> Thanks for your valuable feedback. We will detail the techniques adapted from RoarGraph to make the paper self-contained in a revised manuscript. Below we explain how we adapt the projection technique to RetrievalAttention.
>
> The projection technique aims to construct a graph index by retaining only key vectors, while ensuring that relevant key vectors identified by the query vectors remain closely connected in the graph structure. Therefore, benefiting from the elimitation of query vectors, it reduces the memory footprint and accelerates the search process.
>
> During index construction, we first use query vectors to identify the top-$m$ important key vectors to query vectors with highest inner product, as demonstrated in Figure 4(b) in our paper. After that, it comes to the projection phase to form a graph index.
>
> - The first phase is to connect $m$ key vectors, which are recognized as relevant by a query vector, in the graph structure. This process iterates for each query vector with top-$m$ key vectors. After this process, we can obtain a graph data structure that only has key vectors. However, the connectivity of this graph may not be good enough because some key vectors do not have enough neighbors.
> - The second phase is to enhance the connectivity of the graph structure. For each key vector, we follow the traditional graph index building to perform the nearest neighbor search in the graph to add more connections if one key vector does not have enough neighbors.
> - One of the most important adaption of RetrievalAttention is that, we use Euclidean distance instead of inner product score to evaluate distance between key vectors during the projection. We made this design choice because we found that, in the Q/K vectors from attention mechanism, some key vectors may dominate the neighboring space and have too large inner product score. Other key vectors tend to connect to these vectors, which will have large inner product score, within a limited node degree. This makes other key vectors hard to connect with each other and incurs low connectivity. However, we found that changing it to the pure geometry space metric (Eudclidean distance) can effectively modeling key vectors' relationship, helping establish edges between key vectors successfully.
>
> Although projection technique is introduced by RoarGraph, the technique details regarding the adaptation to RetrievalAttention would enhance the novelty of our paper. We will take this review comment to add more details and further improve the paper.
>
>
> W2. ***"Key design choices, such as the vector projection technique and the static token heuristic, lack comparative ablation studies to justify their selection over other alternatives."***
>
> Thank you for your suggestion. We conducted ablation experiments of the two techniques and make the following analysis.
>
> 1. We first study the effect of the projection technique by comparing RetrievalAttention's vector index with a vector index without projection. Specifically, the index without projection includes query vectors with connections to key vectors that identified as critical by query vectors. Searching on such bi-directional graph structure will require routing between query and key vectors, resulting in excessive accesses.
>
> For the Q/K vectors dumped from different models, Table 1 list the percentage of number of accessed vectors required by the graph index without and with projection for achieving recall@100$\ge$ 0.95. The results show that indexes without projection are inefficient to achieve high accuracy.
>
> **Table 1: The effect of projection in vector index quality.**
> |Model|Recall@100 (w/o projection) | Scanned vectors (%) (w/o projection) | Recall@100 (RetrievalAttention, w/ projection)| Scanned vectors (%) (RetrievalAttention, w/ projection)|
> |---|---|---|---|---|
> |Llama3-8B-Instruct-262k| 0.951 | 13.2% |**0.954**|**1.7%**|
> |Yi-6B-200K| 0.954 | 27.8% |**0.958**|**1.7%**|
> |Yi-9B-200K| 0.956 | 23.4% |**0.954**|**1.6%**|
>
> 2.  We conducted a sensitivity analysis by varying the size of the static pattern using Llama-3-8B on the RULER benchmark (128K). As shown in the Table 2, as the size of the static pattern increases, the accuracy gradually improves and tends to be stable. This is consistent findings in previous works that the number of sink tokens is relatively small [1]. We also observe that the decoding latency of RetrievalAttention remains unchanged, because the static pattern is computed efficiently on the GPU and fully overlapped with index operations on the CPU.
>
> We will include above experiments and analysis in the revised version of paper.
>
> **Table 2: RetrievalAttention's accuracy with different static pattern sizes.**
> |Static pattern size|16+64|32+128|64+256|128+512|
> |-|-|-|-|-|
> |Accuracy on RULER|73.93%|74.15%|74.52%|74.70%|
>
>
> Q1.***"why is a simple IVF often comparable to RetrievalAttention? (e.g. Table 3, Yi-*B)"***
>
> Thank you for the comment. You have pinpointed a crucial trade-off that is important to our work's motivation: the balance between retrieval accuracy and the computational overhead required to achieve it.
>
> A simple IVF can sometimes achieve comparable accuracy because it incurs a significantly higher search cost (scan 30% of all vectors) than RetrievalAttention (scan 1--3% of vectors). Our work aims to achieve strong accuracy while dramatically reducing search overhead. As shown in the Table 7 of Appendix A.8 in the original submission, the index retrieval latency of IVF is approximately 5$\times$ higher than RetrievalAttention, and its end-to-end decoding latency is 2.8$\times$ higher.
>
> We will clarify this in the revised version of this paper.
>
> [1] Efficient Streaming Language Models with Attention Sinks. ICLR 2024

---

> ### Comment · Reviewer_6mEN · 2025-08-06
>
> Thank you for your detailed response. I will keep my positive rating.

---

> > ### Author Response · Authors · 2025-08-07
> >
> > We sincerely appreciate your feedback and the recognition of our work!

---

### Official Review · Reviewer_b7VJ · 2025-07-02

**Clarity:** 3
**Significance:** 3
**Originality:** 2
**Rating:** 5
**Confidence:** 5

**Summary:**

The paper introduces RetrievalAttention - a training-free approach for scaling long-context inference that off-loads most key/value vectors to CPU memory and serves them through an attention-aware ANN index, while keeping only a small “always-useful” window on the GPU. By retrieving roughly 1-3 % of tokens per query and merging the partial results with a FlashAttention-style kernel, it maintains accuracy within two percentage points of full attention but delivers up to an 8× speed-up and enables 128 K–1 M-token contexts on a single 24 GB GPU.

**Questions:**

1. In Appendix A.14, index building takes significant time—have you experimented with GPU-accelerated variants to make it more practical?
2. How long does the offline KNN + projection index construction take for a 1 M-token corpus, and how often must it be rebuilt if the fixed context drifts? Providing wall-clock numbers would clarify practical deployability.
3. How sensitive is performance to the choice of static patterns (e.g., varying the 640-token window)?

**Ethical Concerns:**

["NO or VERY MINOR ethics concerns only"]

**Final Justification:**

The authors clarified the points, I decide to keep my original positive score.

**Limitations:**

Yes - it discusses that the prefill side is not optimized in this work. It would be helpful to include CPU overhead in the limitations as well.

**Paper Formatting Concerns:**

no major issues

**Quality:**

3

**Strengths And Weaknesses:**

Strenghts:
* Strong empirical results of decoding speedup at 128K context length, implying practical usability.
* Novel framing of sparse attention as a cross-distribution MIPS problem and corresponding OOD-aware index design.
* Thorough empirical study on open-source LLMs and three long-context benchmarks and ablations on index recall.


Weaknesses:
* The CPU cost of building the index need more transparency in the main part of the paper
* Related-work section does not include prior work on kNN attention such as Memorizing Transformer (Wu et al., 2022) and Focused Transformer (Tworkowski et al., 2023)

---

> ### Author Rebuttal · Authors · 2025-07-31
>
> We sincerely appreciate the reviewer's constructive feedback and will take the review comments to further improve the paper. We address each review comment below.
>
> W1. ***"The CPU cost of building the index need more transparency in the main part of the paper."***
>
> We thank the reviewer for this comment and agree that the building cost should be presented in the main paper. We will do this in the revised paper.
>
> Q1. **"In Appendix A.14, index building takes significant time—have you experimented with GPU-accelerated variants to make it more practical?"**
>
> Yes, we have tried to integrate a recent GPU-based index building method [1] into RetrievalAttention. Our preliminary experiments shows that it significantly reduces the building time by approximately 3$\times$.
>
> Q2. ***"How long does the offline KNN + projection index construction take for a 1 M-token corpus, and how often must it be rebuilt if the fixed context drifts? Providing wall-clock numbers would clarify practical deployment."***
>
> Thanks for this comment. We evaluated the index building time using CPU for all layers on longer contexts, following the same setting in Appendix A.14.
>
> As detailed in Table 1, index building time increases with context length, because of the increasing size of KV vectors. For the 1M tokens, it takes 9174 seconds. As noted in the response to the first question, GPU-based acceleration can be used to reduce this cost.
>
> **Table 1:** Index building time with different context lengths.
> | Context length | 128K     | 256K     | 512K      | 1M        |
> | -------------- | -------- | -------- | --------- | --------- |
> | Build time     | 844 s    | 1568 s   | 3812 s    | 9174 s    |
>
>
> Regarding the case of fixed context drift, we categorize it into two scenarios.
> 1. If the original context remains unchanged and new content is only appended to the end of the original one, RetrievalAttention is capable of supporting incremental index updates with relatively low cost, by following the update strategy in [2].
> 2. If the content of the fixed context changes, the KV cache would change so we need to rebuild the indexes on the updated KV cache.
>
>
> W2. ***"Related-work section does not include prior work on kNN attention such as Memorizing Transformer (Wu et al., 2022) and Focused Transformer (Tworkowski et al., 2023)"***
>
> We appreciate this comment and agree that a discussion about KNN attention is important to clarify the landscape and positioning of RetrievalAttention.
>
> While Memorizing Transformer and Focused Transformer also leverage kNN-based retrieval, RetrievalAttention differs from them in the design goals and application scenarios. Memorizing Transformer and Focused Transformer focus on extending the context window and sometimes requires model training to mitigate the distraction issue. In comparison, RetrievalAttention builds upon existing models with long-context capacity and aims to improve the long-context inference efficiency.
>
> We will incorporate these related works into the revised revision of our paper.
>
>
> Q3. ***"How sensitive is performance to the choice of static patterns (e.g., varying the 640-token window)?"***
>
> Thanks for this comment. We conducted a sensitivity analysis by varying the size of the static pattern using Llama-3-8B on the RULER benchmark (128K).
>
> As shown in the Table 2, as the size of the static pattern increases, the accuracy gradually improves and tends to be stable. This is consistent findings in previous works that the number of sink tokens is relatively small [3]. We also observe that the decoding latency of RetrievalAttention remains unchanged, because the static pattern is computed efficiently on the GPU and fully overlapped with index operations on the CPU.
>
> We will include above experiments and analysis in the revised version of paper.
>
> **Table 2: RetrievalAttention's accuracy with different static pattern sizes.**
> |Static pattern size|16+64|32+128|64+256|128+512|
> |-|-|-|-|-|
> |Accuracy on RULER|73.93%|74.15%|74.52%|74.70%|
>
>
> [1] ParaGraph: Accelerating Graph Indexing through GPU-CPU Parallel Processing for Efficient Cross-modal ANNS, DaMoN@SIGMOD 2025.
>
> [2] RoarGraph: A Projected Bipartite Graph for Efficient Cross-Modal Approximate Nearest Neighbor Search，VLDB 2024
>
> [3] Efficient Streaming Language Models with Attention Sinks. ICLR 2024

---

### Official Review · Reviewer_efRu · 2025-07-05

**Clarity:** 2
**Significance:** 3
**Originality:** 3
**Rating:** 4
**Confidence:** 3

**Summary:**

This paper introduces RetrievalAttention, a framework that accelerates long-context inference by pairing a retrieval‐based attention mechanism with a hybrid CPU–GPU KV-cache design. By building an attention-aware index over a large, mostly fixed prefix and fetching only the most relevant keys at run time, the approach reduces GPU memory usage and latency while preserving model quality. Experiments across several models and benchmarks show substantial speed-ups with minimal accuracy loss.

**Questions:**

1. Please report wall-clock build time, peak RAM, and index size for 128 K / 1 M tokens. How would you update the index if new tokens are appended or removed?

2. Can RetrievalAttention operate when all tokens arrive online (no fixed prefix)? What is the end-to-end latency if the index must grow on the fly?

3. Some attention-free approaches also claim large-context support. Could you add empirical results or explain why comparison is infeasible?

**Ethical Concerns:**

["NO or VERY MINOR ethics concerns only"]

**Final Justification:**

The authors clarified several design choices and provided further comparisons, which addressed my main initial concerns.

The rebuttal improved the presentation of the method, especially regarding how retrieval integrates with attention. Additional experiments strengthened the empirical evaluation, showing efficiency gains in long-context settings. While the improvements are not dramatic in all cases, they are consistent and relevant.

Given the clearer positioning and solid empirical validation, I find the contribution sufficient for acceptance.

**Limitations:**

yes

**Quality:**

3

**Strengths And Weaknesses:**

**Strenghts**:

1. This paper systematically quantify how ANN search degrades when query and key distributions diverge, then designs an index that is query-aware by construction.

2. The CPU–GPU split plus attention-aware index enables commodity GPUs to serve 100 K-scale contexts without quantization, delivering tangible memory and latency gains.

**Weakness**:

1. It seems the paper omits concrete numbers for build time, peak RAM, and index size at 128 K–1 M tokens, making deployability hard to judge.

2. The evaluation parts did not include attention-free compression methods into comparison.

3. The method assumes a largely fixed prefix. For interactive or rapidly changing inputs (chat, code completion) the indexing cost and latency overhead are unclear and untested.

---

> ### Author Rebuttal · Authors · 2025-07-31
>
> We sincerely appreciate the reviewer's constructive feedback. As recognized by the reviewer, our key contribution lies in the novel application of attention-aware vector indexes to long-context LLM inference, coupled with systems designs for both high accuracy and efficiency. We will take the review comments to further improve the paper. We address each review comment below.
>
> W1. ***"It seems the paper omits concrete numbers for build time, peak RAM, and index size at 128 K–1 M tokens, making deployability hard to judge."***
>
> Q1. ***"Please report wall-clock build time, peak RAM, and index size for 128 K / 1 M tokens."***
>
> Thank you for this comment. We provide the requested statistics for index build time, peak RAM consumption, and index size when constructing indexes on vector data derived from Llama-3-8B. The index is built layer by layer, from the initial to the final layer. Therefore, peak memory occurs during the index building of the final model layer, comprising the KV vectors, the cumulative index size from all preceding layers, and intermediate states of the single-layer index build.
>
> As detailed in Table 1, all three metrics increase with context length due to the growing size of the KV vectors. The index building time is acceptable, given that this process is performed offline on CPUs. Regarding peak RAM and index size, the observed memory footprint remains manageable on commodity hardware.
>
> Index building and memory footprint can be optimized with more advanced techniques. To further reduce build time, we conducted preliminary experiments using a recent GPU-based index building method [1] and observed a significant speedup of approximately 3$\times$. To optimize memory consumption, we identify vector quantization and offloading data to storage devices as promising directions for future research.
>
>
> **Table 1:** Statistics of the index building.
> | Context length | 128K     | 256K     | 512K      | 1M        |
> | -------------- | -------- | -------- | --------- | --------- |
> | Build time     | 844 s    | 1568 s   | 3812 s    | 9174 s    |
> | Peak RAM       | 28.8 GiB | 56.6 GiB | 109.5 GiB | 208.8 GiB |
> | Index size     | 9.7 GiB  | 19.4 GiB | 39.1 GiB  | 77.9 GiB  |
>
>
> W2. ***"The evaluation parts did not include attention-free compression methods into comparison."***
>
> Q3. ***"Some attention-free approaches also claim large-context support. Could you add empirical results or explain why comparison is infeasible?"***
>
> We appreciate this comment. Since the reviewer did not specify which “attention-free methods” they referred to, we assume the term denotes models that avoid the use of attention mechanisms, such as Mamba [2] and RWKV [3].
>
> Initially, our focus was on comparing RetrievalAttention with other attention-based long-context methods, as our approach directly enhances the transformer's attention efficiency by leveraging sparsity. However, we acknowledge the growing interest in attention-free architectures.
>
> While these attention-free models offer compelling theoretical advantages, recent comprehensive long-context benchmarks, such as RULER [4], have indicated that their accuracy on diverse long-context tasks still lags significantly behind attention-based transformer architectures. Given that RetrievalAttention aims to achieve accuracy comparable to full attention models while significantly reducing computational overhead, it is expected to outperform these attention-free methods on long-context tasks.
>
> To empirically validate this, we conducted an additional comparison with a popular attention-free model, Falcon-Mamba-7B-Instruct, on the RULER benchmark. As shown in Table 2, even at relatively modest context lengths (e.g., 16K), Mamba-7B's accuracy is substantially lower than that of both attention-based models (represented by Llama-3-8B) and RetrievalAttention. The result supports our initial reasoning and highlights that while attention-free methods are an active area of research for long-context modeling, they have not yet demonstrated competitive performance for practical deployment.
>
> We plan to include a discussion of this aspect in the revised version of the paper, to clarify the competitive landscape and positioning of RetrievalAttention.
>
> **Table 2:** Accuracy of different models and methods on the RULER benchmark.
> |Context length|Falcon-Mamba-7B|Llama-3-8B|RetrievalAttention|
> |-|-|-|-|
> |16K|21.79%|89.27%|86.80%|
> |32K|5.66%|85.11%|84.78%|
>
>
> W3. ***"The method assumes a largely fixed prefix. For interactive or rapidly changing inputs (chat, code completion) the indexing cost and latency overhead are unclear and untested."***
>
> Q2. ***"Can RetrievalAttention operate when all tokens arrive online (no fixed prefix)? What is the end-to-end latency if the index must grow on the fly?"***
>
>
> We thank the reviewer for this comment and would like to clarify the scope of RetrievalAttention and potential extensions to fully dynamic scenarios.
>
> We acknowledge that RetrievalAttention currently focus on scenarios with a largely fixed prefix. As explicitly stated in our paper, this design choice addresses a critical need in many practical use cases, such as document question answering, where the context (e.g., a document) remains consistent while queries change.
>
> We agree with the reviewer that extending RetrievalAttention to handle rapidly changing inputs is an important direction. When all tokens arrive online, RetrievalAttention requires building indexes on the fly during the prefill phase. This dynamic index construction adds overhead and can impact the Time-to-First-Token (TTFT). For a 128K context, as shown in the Table 1 above, the TTFT would be approximately 884 seconds without specialized acceleration. GPU-based acceleration [1] can be used to reduce such overhead to approximately 300 seconds. We believe more advanced acceleration deserves exploration to minimize the indexing cost for more dynamic use cases.
>
> Once the index is built, the subsequent token-by-token decoding latency of RetrievalAttention remains highly efficient, consistent with the results reported in our paper for fixed-prefix scenarios.
>
>
> Q1. ***" How would you update the index if new tokens are appended or removed?"***
>
> Thanks for this comment. Our current design prioritizes index stability to enable index reuse across user requests. Therefore, for newly generated tokens, we choose to append them to the GPU memory as a static pattern, avoiding incremental index updates. This is a deliberate design choice to not change the index of the shared prefix (e.g., documents). Otherwise, indexes require handling deletion or reconstruction, impacting the serving latency. For example, if a long document is used as a shared prefix, inserting tokens at runtime would change the index, making it incompatible with other user requests using the same prefix context.
>
> While our design favors static appending for shared prefixes, we clarify that RetrievalAttention is capable of supporting incremental index updates, by adopting the update strategy in [5]. To validate this, we conducted an experiment using a 128K summary generation task with Llama3-8B, generating approximately 1.2K tokens. We inserted all generated tokens into the index and evaluated the recall performance. Our findings demonstrate that RetrievalAttention's index robustly maintains high recall quality even when new key vectors are continuously inserted. Moreover, by conducting incremental inserts, it takes a 10% latency increase in per token generation, making 0.107s per token to 0.118s per token.
>
> [1] ParaGraph: Accelerating Graph Indexing through GPU-CPU Parallel Processing for Efficient Cross-modal ANNS, DaMoN@SIGMDO 25
>
> [2] Mamba: Linear-Time Sequence Modeling with Selective State Spaces, 2023.
>
> [3] RWKV: Reinventing RNNs for the Transformer Era, 2023
>
> [4] RULER: What’s the Real Context Size of Your Long-Context Language Models? COLM 2024
>
> [5] RoarGraph: A Projected Bipartite Graph for Efficient Cross-Modal Approximate Nearest Neighbor Search，VLDB 2024

---

> > ### Comment · Reviewer_efRu · 2025-08-08
> >
> > Thank you for your detailed response. My concerns have been largely addressed.
> > For attention-free methods, I'm referring to quantization or model compression methods. But comparing with Mamba and RWKV is complementary. The current positive score reflects my evaluation of the paper.

---

> > > ### Author Response · Authors · 2025-08-08
> > >
> > > Thanks for your effort in reviewing our paper and for your recognition!
> > > KV cache quantization methods focus on the dimension level, while model compression techniques target the model weights. These approaches are orthogonal to RetrievalAttention’s optimizations at the sequence level. We will include a discussion of these methods in the future version of the paper. Your suggestions have greatly helped us improve the work.

---

### Note · Authors · 2025-08-13

Dear Area Chairs and Reviewers,

We are grateful to the reviewers for their constructive comments, which have helped us refine our work.

The reviewers have recognized key contributions of our paper, including the novelty of applying vector indexes to long-context LLM inference, systematic analysis of out-of-distribution behaviors, the superior performance of attention-aware indexes and system designs, and comprehensive evaluations.

**[Novelty]**
* b7VJ: "Novel framing of sparse attention as a cross-distribution MIPS problem..."
* krY7: "It's innovative to use the vector-search methods for the KV cache retrieval..."
* efRu: "...systematically quantify how ANN search degrades when query and key distributions diverge..."
* 6mEN: "... a clear and compelling motivating analysis of the OOD problem."

**[Superior Systems Efficiency]**
Reviewers acknowledged that our system design is "simple and effective" (6mEN) and delivers "tangible memory and latency gains" (efRu). Reviewer b7VJ also recognized that our "Strong empirical results...implying practical usability."

**[Comprehensiveness]**
Reviewers recognized that our evaluation is "solid" and "thorough".

During the rebuttal period, we carefully addressed all feedback and conducted supplementary experiments for clarification. We concisely summarize our responses below:

**[Experiments]**
We have added following experiment results:
* Reported index building time, peak RAM usage, and index size for larger context lengths, ensuring practical applicability of our method (efRu W1, b7VJ Q2).
* Evaluated GPU-accelerated index building methods (b7VJ Q1).
* Ablation study of static pattern sizes (b7VJ Q3, 6mEN W2).
* Ablation study of the projection technique (6mEN W2).

**[Related Work]**
We have expanded the discussion of related works including non-transformer models (efRu W2) and kNN attention methods (b7VJ W2). We also clarified quantization methods are orthogonal to us (efRu W2).

**[More Explanations]**
We added more explanations as follows:
* Clarified the potential extensions to fully dynamic scenarios and the handling of new tokens (efRu W3, Q1).
* Detailed explanations of adaptations of the projection techniques (6mEN W1).
* Clarified the results and limitations of IVF (6mEN Q1).
* Clarified full attention setup and the superlinear results (krY7 Q1).
* Explained the scenarios with batch processing (krY7 Q2).

We sincerely hope our detailed responses have addressed concerns from reviewers.

Best,

Authors

---

### Decision · Program_Chairs · 2025-09-17

**Decision:**

Accept (poster)

**Comment:**

This paper proposes an approach (RetrivalAttention) which offloads the KV cache to the CPU and builds a vector-search index that helps nearest neighbor search and retrieval during the decoding process. Empirical results show that the proposed solution can run long context inference on a consumer-grade GPU with limited memory with a small quality drop. Overall, reviewers pointed out as strengths the innovative use of vector-search methods for KV cache retrieval and the solid evaluation, and as weaknesses the lack of detail about the overheads of creating the index and a lack of clarity about the "full attention" baseline. The rebuttal clarified the second point and promised to provide details on the index creation. Overall this seems a solid paper. I urge the authors to follow the suggestions from the reviewers.